

# Clinical applicability of the Feline Grimace Scale: real-time versus image scoring and the influence of sedation and surgery

Marina C. Evangelista[1], Javier Benito[1], Beatriz P. Monteiro[1], Ryota Watanabe[1], Graeme M. Doodnaught[1], Daniel S.J. Pang[1,2] and Paulo V. Steagall[1]

[1] Department of Clinical Sciences, Faculty of Veterinary Medicine, Université de Montréal, St-Hyacinthe, Quebec, Canada
[2] Veterinary Clinical and Diagnostic Sciences, Faculty of Veterinary Medicine, University of Calgary, Calgary, Alberta, Canada

Corresponding author
Paulo V. Steagall,
paulo.steagall@umontreal.ca

## ABSTRACT

**Background**. The Feline Grimace Scale (FGS) is a facial expression-based scoring system for acute pain assessment in cats with reported validity using image assessment. The aims of this study were to investigate the clinical applicability of the FGS in real-time when compared with image assessment, and to evaluate the influence of sedation and surgery on FGS scores in cats.

**Methods**. Sixty-five female cats (age: $1.37 \pm 0.9$ years and body weight: $2.85 \pm 0.76$ kg) were included in a prospective, randomized, clinical trial. Cats were sedated with intramuscular acepromazine and buprenorphine. Following induction with propofol, anesthesia was maintained with isoflurane and cats underwent ovariohysterectomy (OVH). Pain was evaluated at baseline, 15 min after sedation, and at 0.5, 1, 2, 3, 4, 6, 8, 12 and 24 h after extubation using the FGS in real-time (FGS-RT). Cats were video-recorded simultaneously at baseline, 15 min after sedation, and at 2, 6, 12, and 24 h after extubation for subsequent image assessment (FGS-IMG), which was performed six months later by the same observer. The agreement between FGS-RT and FGS-IMG scores was calculated using the Bland & Altman method for repeated measures. The effects of sedation (baseline versus 15 min) and OVH (baseline versus 24 h) were assessed using linear mixed models. Responsiveness to the administration of rescue analgesia (FGS scores before versus one hour after) was assessed using paired $t$-tests.

**Results**. Minimal bias ($-0.057$) and narrow limits of agreement ($-0.351$ to $0.237$) were observed between the FGS-IMG and FGS-RT. Scores at baseline (FGS-RT: $0.16 \pm 0.13$ and FGS-IMG: $0.14 \pm 0.13$) were not different after sedation (FGS-RT: $0.2 \pm 0.15$, $p = 0.39$ and FGS-IMG: $0.16 \pm 0.15$, $p = 0.99$) nor at 24 h after extubation (FGS-RT: $0.16 \pm 0.12$, $p = 0.99$ and FGS-IMG: $0.12 \pm 0.12$, $p = 0.96$). Thirteen cats required rescue analgesia; their FGS scores were lower one hour after analgesic administration (FGS-RT: $0.21 \pm 0.18$ and FGS-IMG: $0.18 \pm 0.17$) than before (FGS-RT: $0.47 \pm 0.24$, $p = 0.0005$ and FGS-IMG: $0.45 \pm 0.19$, $p = 0.015$).

**Conclusions**. Real-time assessment slightly overestimates image scoring; however, with minimal clinical impact. Sedation with acepromazine-buprenorphine and ovariohysterectomy using a balanced anesthetic protocol did not influence the FGS scores. Responsiveness to analgesic administration was observed with both the FGS-RT and FGS-IMG.

## INTRODUCTION

Despite the advent of scoring systems and assessment tools, pain is still under-recognized and under-treated in cats worldwide (*Lorena et al., 2014*; *Hunt et al., 2015*; *Reimann et al., 2017*; *Morales-Vallecilla et al., 2019*). Facial expressions are considered a reliable method of pain assessment in non-verbal humans and other mammals (*Prkachin, 2009*; *Langford et al., 2010*; *Sotocinal et al., 2011*; *Dalla Costa et al., 2014*; *Finka et al., 2019*).

Linear distances between specific facial landmarks and the quantification of changes in facial shape for the study of pain in cats have been published (*Holden et al., 2014*; *Finka et al., 2019*). Recently, a facial expression-based scoring system has been proposed for assessing acute pain in cats, namely the Feline Grimace Scale (FGS). This instrument has been developed and validated in cats with different sources and intensities of pain produced by naturally-occurring conditions; additionally, its reliability, criterion and construct validity (including responsiveness) were assessed using image scoring (*Evangelista et al., 2019*).

The standard methodology for assessing facial expressions of pain using grimace scales usually relies on image scoring (*Langford et al., 2010*; *Sotocinal et al., 2011*; *Dalla Costa et al., 2014*). The images are commonly screenshots obtained from videos which represents a time-consuming procedure that takes place weeks or months after video-recordings. In clinical practice, pain must be promptly assessed to ensure appropriate and immediate treatment. Real-time scoring has been described using the Mouse Grimace Scale (MGS; *Miller & Leach, 2015*) and the Rat Grimace Scale (RGS; *Leung, Zhang & Pang, 2016*), but not the FGS.

Elective sterilization of female companion animals (i.e., ovariohysterectomy, OVH) is one of the most common surgical procedures performed in veterinary medicine. The administration of sedatives, anesthetics and analgesics are required in the perioperative setting and drugs such as ketamine have been shown to produce confounding effects by increasing psychomotor scores with feline pain scales (*Buisman et al., 2016*). Indeed, the influence of anesthesia, surgery and opioids (i.e., buprenorphine) on grimace scale scores have been demonstrated in mice and rats (*Miller et al., 2015*; *Leung, Zhang & Pang, 2016*). Changes in facial shape have been reported in cats after OVH (*Finka et al., 2019*). It is not known how sedation and OVH itself may affect FGS scores, more specifically, and if these represent potential limitations in the application of the FGS in clinical practice.

The objectives of this study were: (1) to investigate whether the FGS could be successfully implemented in a clinical setting in cats undergoing OVH, enabling real-time pain assessment; (2) to evaluate the influence of sedation and OVH on the FGS; and (3) to reassess the responsiveness to analgesic treatment of the FGS in real-time. Our hypotheses were: (1) FGS scores obtained in real-time and by image assessment would be comparable when evaluated by the same observer; (2) FGS scores would not change after sedation with acepromazine and buprenorphine or at 24 h after extubation when

compared with baseline values; and (3) the FGS scores in real-time would decrease after analgesic treatment.

## MATERIALS & METHODS

### Animals

Eighty-one adult female domestic cats from local shelters were admitted to the veterinary teaching hospital (*Centre Hospitalier Universitaire Vétérinaire*) of the Faculty of Veterinary Medicine, Université de Montréal for elective OVH between June and October 2018. The study was designed to evaluate the analgesic efficacy of intraperitoneal administration of bupivacaine alone or in combination with dexmedetomidine (*Benito et al., 2019*). The current study occurred in parallel to evaluate changes in the FGS. The protocol was approved by the local animal care committee, *Comité d'éthique de l'utilisation des animaux—Université de Montréal* (protocol number 18-Rech-1825). Written informed consent for participation in the study was obtained for each patient.

Inclusion criteria included healthy cats based on physical examination and hematology. Exclusion criteria included cats presenting with cardiac arrhythmias, body condition score >7 or <3 on a scale from 1 to 9, anemia (hematocrit < 25%), hypoproteinemia (total protein <5.9 g/dL), cats with high pain scores at presentation (see pain assessment), or any clinical signs of diseases during physical examination such as upper respiratory tract disease and conjunctivitis. Cats were admitted the day before the procedures and discharged 24 h after the surgery. They were housed individually in adjacent cages in a cat-only ward with free access to water and food, and a litter box. Environmental enrichment included hanging toys, blankets and a cardboard box that cats could use as a hiding spot or as an elevated surface.

### Anesthesia and surgery

Anesthetic and surgical procedures are described thoroughly elsewhere (*Benito et al., 2019*). In brief, food was withheld for 8–12 h before general anesthesia; cats were premedicated with acepromazine (0.02 mg/kg) and buprenorphine (0.02 mg/kg) intramuscularly; anesthesia was induced with propofol intravenously to allow endotracheal intubation (4–6 mg/kg) and maintained with isoflurane in oxygen. Cats were randomly assigned to receive one of the two following treatments: bupivacaine (2 mg/kg) or bupivacaine (same dose) with dexmedetomidine (1 μg/kg). All cats received an intraperitoneal administration of either bupivacaine alone or in combination with dexmedetomidine intraoperatively. The intraperitoneal sterile solution was instilled (splashed) before OVH by the veterinary surgeon, over the right and left ovarian pedicles, and the caudal aspect of the uterine body in three equal volumes. Ovariohysterectomy was performed approximately one minute after intraperitoneal administration using the pedicle tie technique. All cats were discharged the day after surgery and returned to their respective shelters for adoption.

### Sedation and pain assessment

Sedation was assessed using a 5-point simple descriptive scale, where 0 = no sedation, 1 = able to stand but is wobbly; 2 = in sternal recumbency; 3 = can lift its head; 4 = fast asleep/no response to hand clap (adapted from *Steagall et al., 2009*).

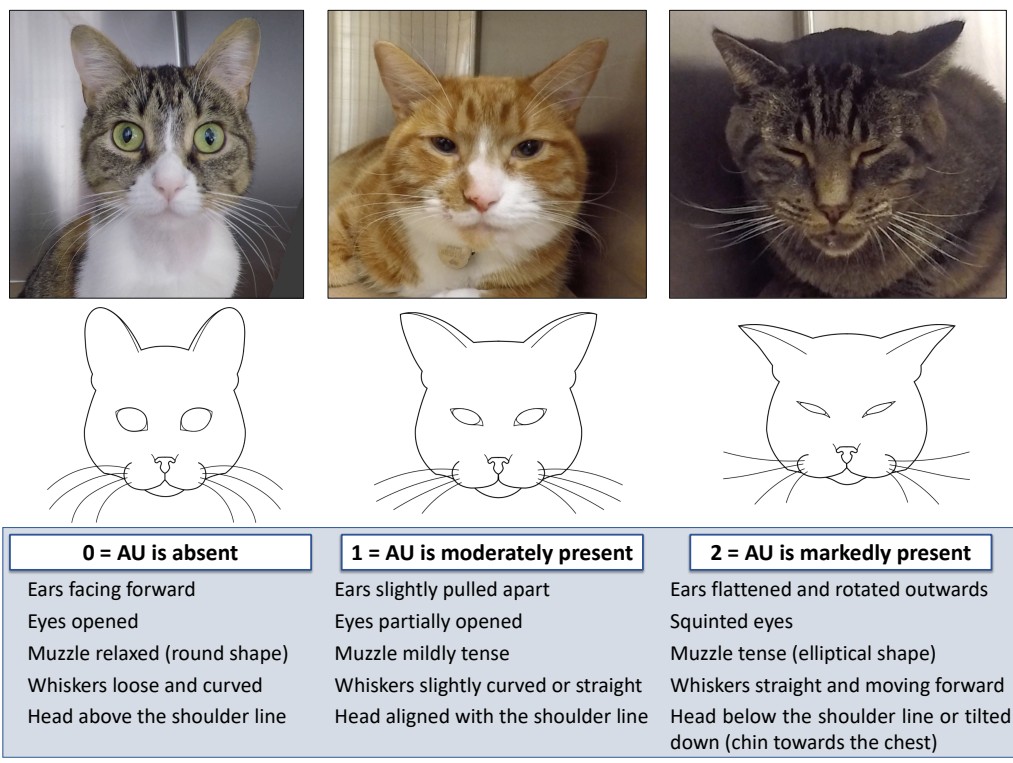

| 0 = AU is absent | 1 = AU is moderately present | 2 = AU is markedly present |
|---|---|---|
| Ears facing forward | Ears slightly pulled apart | Ears flattened and rotated outwards |
| Eyes opened | Eyes partially opened | Squinted eyes |
| Muzzle relaxed (round shape) | Muzzle mildly tense | Muzzle tense (elliptical shape) |
| Whiskers loose and curved | Whiskers slightly curved or straight | Whiskers straight and moving forward |
| Head above the shoulder line | Head aligned with the shoulder line | Head below the shoulder line or tilted down (chin towards the chest) |

**Figure 1** **Illustration and description of the Feline Grimace Scale.** The Feline Grimace Scale is composed of five action units (ear position, orbital tightening, muzzle tension, whiskers change and head position), each one is scored from 0 to 2 (0 = action unit is absent; 1 = moderate appearance of the action unit, or uncertainty over its presence; and 2 = obvious appearance of the action unit).

Pain was evaluated using the short-form UNESP-Botucatu composite pain scale (SF-UBCPS) and the FGS. The SF-UBCPS is a novel pain scoring system consisting of 4 items (each item is scored from 0 to 3 adding up to a maximum score of 12 points) to evaluate the cats' posture, activity, attitude and reaction to touch and palpation of a painful site. This scale has been validated using video-based assessments (P Steagall, 2019, unpublished data). Cats with a baseline pain score of $\geq 2$ using the SF-UBCPS were not included in the study. High baseline scores could indicate a mild degree of pain before the surgery, or an influence of the cat's demeanor (shy, fearful or feral) (*Buisman et al., 2017*). The FGS consists of five action units (ear position, orbital tightening, muzzle tension, whiskers change and head position), each one is scored from 0 to 2 (0 = action unit is absent; 1 = moderate appearance of the action unit, or uncertainty over its presence; and 2 = obvious appearance of the action unit) (Fig. 1). If one or more action units were not visible, the observer had the option of marking "not possible to score" and the weight was redistributed to the other action units. A total FGS score was calculated by the sum of the scores of the action units divided by the total possible score, excluding those marked as not possible to score (e.g., 4/8 = 0.5). Images with more than 2 "not possible to score" were excluded from final analysis.

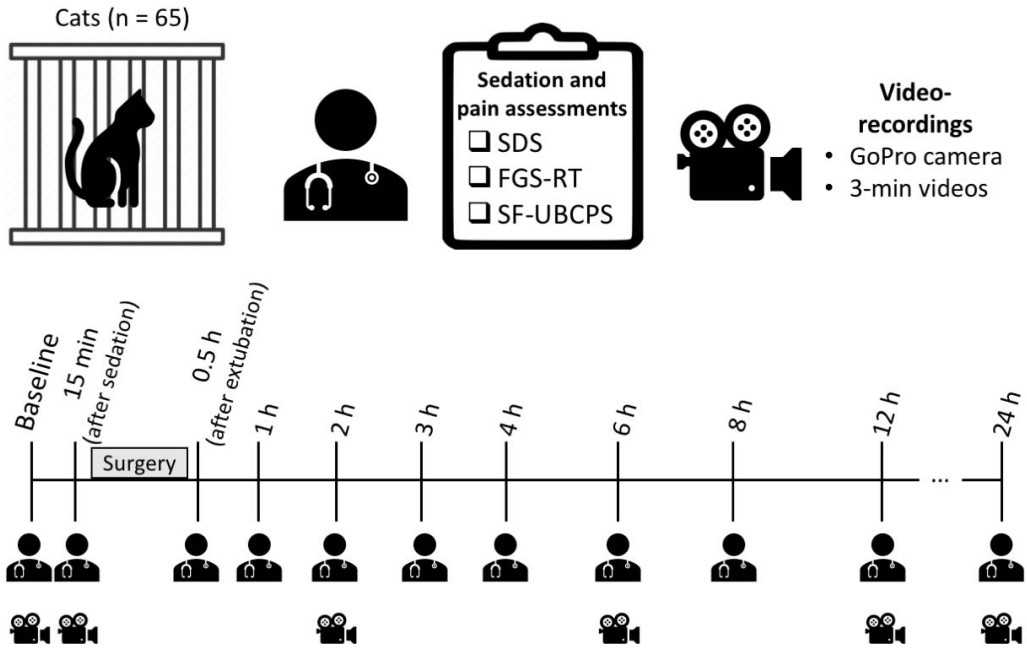

**Figure 2  Timeline of the study and the time-points for sedation and pain assessments in real-time and video-recordings.** Sedation was evaluated with a 5-point simple descriptive scale (SDS). Pain was evaluated using the Feline Grimace Scale (FGS) and the short-form UNESP-Botucatu composite pain scale (SF-UBCPS). FGS scores were obtained in real time (FGS-RT) during three minutes of observation of the cats undisturbed. Three-minute videos were recorded simultaneously to the FGS-RT assessment using a GoPro camera placed between the cage bars.

Sedation and pain assessments were performed by one observer (MCE) at baseline (morning before the surgery), 15 min after premedication, and at 0.5, 1, 2, 3, 4, 6, 8, 12 and 24 h after extubation. Sedation was evaluated before pain assessment. The FGS scores were obtained in real time (FGS-RT) using instantaneous sampling during three minutes of observation without interacting with the cat. A stopwatch was set to control the time, and at the end of each minute (first, second and third minutes), one score per action unit was assigned (representing the whole minute of observation) and a total FGS-RT score was obtained. The average of those three scores was considered the FGS-RT score for that time-point [i.e., $(0.5 + 0.4 + 0.5)/3 = 0.47$]. Three-minute videos were recorded simultaneously during the FGS-RT assessment using a high-definition video camera (GoPro Hero 5, GoPro Inc., San Mateo, CA, USA) at the following time-points: baseline, 15 min after premedication, and at 2, 6, 12 and 24 h after extubation (Fig. 2). The camera was placed between the cage bars at the level of the cats' eyes and set to record at 60 frames per second and a medium angle of view. Following the three-minute recording and FGS-RT scoring, the observer proceeded with the interaction and palpation of the wound/painful area to complete the SF-UBCPS.

Rescue analgesia was provided with buprenorphine 0.02 mg/kg intravenously and meloxicam 0.2 mg/kg subcutaneously when SF-UBCPS $\geq 4$ (the analgesic intervention threshold pre-determined for this scale). Additional rescue analgesia was administered

if needed, with the same dose of buprenorphine. If a cat required the administration of rescue analgesia, additional videos (along with the pain evaluation using the FGS-RT and SF-UBCPS) were recorded before and one hour after its administration. If a cat did not require the administration of rescue analgesia before the 12 h time-point, meloxicam was administered (same dose and route of administration as previously described for rescue analgesia). The choice of administering meloxicam at 12 h for the cats that did not require rescue analgesia beforehand is explained by the time gap until the 24 h evaluation and we wanted to make sure cats were not painful overnight.

## Video handling, image selection and image scoring

The videos were downloaded into a computer and renamed after a random sequence of numbers (http://www.random.org). Videos were not recorded postoperatively if the cats had been previously spayed. In these cases, only videos recorded at baseline and 15 min after sedation were considered. Any identification of the cats' names or time-points were deleted for blinding purposes. Screenshots were obtained from video-recordings whenever the cat was facing the camera, but not when they were sleeping, grooming or vocalizing. The resulting images were identified as belonging to the first, second or third minute of the video. From each three-minute video, the single best image per minute was selected for later scoring. Images were not selected if quality was poor (i.e., blurred or dark images or when the cats' faces were partially/completely not visible).

The same observer (MCE) evaluated pain in real-time using the FGS followed by the SF-UBCPS, and scored the images using the FGS (FGS-IMG) in a blinded and randomized order, six months after the experimental study. The average score of the three images (each from the first, second and third minutes) was considered the final FGS-IMG score for that time-point.

## Statistical analysis

Statistical analyses were performed with SAS v.9.3 (SAS Institute, Cary, NC, USA) and GraphPad Prism 7 (GraphPad software, San Diego, CA, USA).

The agreement between FGS-RT and FGS-IMG scores was assessed per minute of observation and for the average of three minutes using the Bland & Altman's method for repeated measures (*Bland & Altman, 2007*). This method describes the agreement between two quantitative measurements by evaluating a bias between the mean differences, and estimating an agreement interval, within which 95% of the differences fall.

Sedation scores (ordinal variable) over time were compared with baseline using a non-parametric approach (Friedman test and Dunn's multiple comparisons test). Sedation scores are presented as median (range).

Linear mixed models with individuals as random effect and time as fixed effect were used to evaluate the time course of the FGS-RT, and the influence of sedation (baseline versus 15 min after premedication) and OVH (baseline versus 24 h after extubation) on both total FGS-RT and FGS-IMG scores (considered continuous variables). Scores obtained after the administration of rescue analgesia were not included in this analysis. FGS-IMG and FGS-RT scores are presented as mean ± standard deviation (SD). Only the

FGS-RT scores were used for the time course evaluation because scores from all cats and all time-points were available. FGS-IMG scores were obtained at fewer time-points and they were sometimes missing due to the lack of acceptable images from some videos.

The Cochran-Mantel-Haenszel test for ordinal scores and repeated measures was used to assess the influence of sedation and OVH on each individual action unit (scores 0, 1 or 2). This approach takes the order into account, thus representing a more specific test and offering great statistical power.

Responsiveness to rescue analgesia (i.e., construct validity) was assessed by comparing the scores before and one hour after the administration of buprenorphine and meloxicam. In this case, after confirming the normality of data distribution using the Shapiro Wilk test, FGS-RT and FGS-IMG scores were compared using paired t-tests. The SF-UBCPS scores (ordinal variable and not normally distributed) were compared using the Wilcoxon test. SF-UBCPS scores are presented as median (range).

$P$ values were adjusted according to the number of comparisons for each analysis. Bonferroni-corrected values of $p < 0.05$ were considered significant.

# RESULTS

Sixty-five mixed-breed cats met the inclusion criteria (age: $1.37 \pm 0.9$ years and body weight: $2.85 \pm 0.76$ kg; domestic short-haired: $n = 52$ and domestic long-haired cats: $n = 13$).

Sixteen cats were excluded because of upper respiratory tract disease ($n = 4$), conjunctivitis ($n = 1$), facial hemiparalysis ($n = 1$), anemia and hypoproteinemia ($n = 1$), body condition score >7/9 ($n = 1$) and <3/9 ($n = 1$) or high baseline SF-UBCPS scores ($n = 7$).

A total of 394 videos and 1373 still images were recorded and obtained, respectively. The median (range) number of selected images per cat was 6 (0–20). The maximum number of possible still images selected per cat would be 18 if this cat did not require rescue analgesia, considering that it could be possible to select one image from each minute for each video of 3 min recorded at the six planned time-points (baseline, 15 min, 2 h, 6 h, 12 h and 24) ($3 \times 6 = 18$). Following the same reasoning, if the cat required rescue (i.e., at 4h), two additional videos would be recorded (at 4 h and 5h) and the maximum number of possible images would be 20 ($18 + 2 = 20$).

The total number of images used for scoring was 540. It was not possible to obtain any image from 160 videos. The selected images ($n = 540$) were scored in four sessions of three hours maximum (135 images per day, every other day) to avoid observer fatigue.

## Agreement FGS-RT and FGS-IMG

Minimal bias and narrow limits of agreement (LOA) were observed between the FGS-RT and FGS-IMG scores. The agreement was calculated for each minute of observation (Figs. 3A–3C) and for the average of three minutes (final score per time-point; Fig. 3D). The FGS-RT scores overestimated FGS-IMG scores (Bias$_{\text{1st min}} = -0.078$, Bias$_{\text{2nd min}} = -0.054$, Bias$_{\text{3rd min}} = -0.050$ and Bias$_{\text{final score}} = -0.057$).

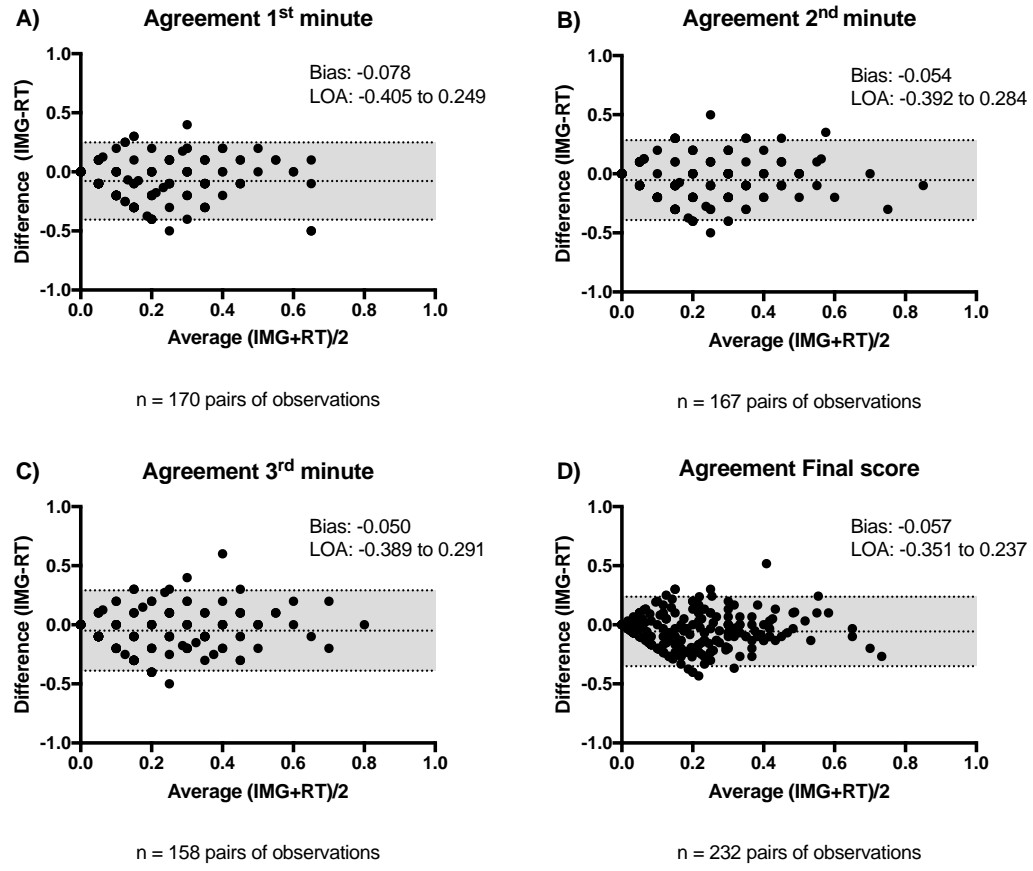

**Figure 3** **Agreement between Feline Grimace Scale scores obtained in real-time (FGS-RT) and by image assessment (FGS-IMG).** Bland & Altman's plots showing the bias and limits of agreement (LOA) between FGS-RT and FGS-IMG for the (A) first minute of observation; (B) second minute of observation; (C) third minute of observation and (D) final score (average of the three minutes). Numbers of pairs of scores used for each analysis are indicated below the charts.

## Sedation scores

Sedation scores were significantly higher than baseline at 15 min after sedation ($p = 0.012$) and 0.5 h after extubation ($p < 0.001$) (Table 1).

## Time course and influence of sedation and OVH on FGS scores

In comparison with baseline, mean ± SD FGS-RT scores were not different at 15 min after sedation, nor at 0.5 h and 24 h after extubation. The FGS-RT scores were significantly higher than baseline from 1 to 12 h after extubation (Table 2 and Fig. 4).

In comparison with baseline, mean ± SD FGS-IMG scores ($0.14 \pm 0.13$) were not different at 15 min after sedation ($0.16 \pm 0.15$, mean difference $= 0.028$, 95% CI of the differences: $-0.0298$ to $0.0864$, $p = 0.9911$) nor at 24 h after extubation ($0.12 \pm 0.12$, mean difference $= -0.033$, 95% CI of the differences: $-0.1$ to $0.0339$, $p = 0.9569$).

**Table 1  Sedation scores obtained over time using a 5-point simple descriptive scale (adapted from *Steagall et al., 2009*).**

| Time-point | Sedation scores | | Median of the differences | 95% CI of the differences | *p* value |
| --- | --- | --- | --- | --- | --- |
| | Median (range) | Interquartile range | | | |
| Baseline | 0 (0–2) | [0–0] | – | – | – |
| 15 min | 1 (0–2)* | [0–2] | 0 | 0 to 1 | 0.012 |
| 0.5 h | 1 (0–3)* | [1–1] | 1 | 1 to 1 | <0.001 |
| 1 h | 1 (0–3) | [0–2] | 0 | 0 to 1 | 0.053 |
| 2 h | 1 (0–3) | [0–2] | 0 | 0 to 1 | 0.051 |
| 3 h | 0 (0–3) | [0–2] | 0 | 0 to 0 | 0.161 |
| 4 h | 0 (0–3) | [0–2] | 0 | 0 to 0 | 0.155 |
| 6 h | 0 (0–3) | [0–2] | 0 | 0 to 0 | >0.999 |
| 8 h | 0 (0–3) | [0–2] | 0 | 0 to 0 | >0.999 |
| 12 h | 0 (0–3) | [0–0] | 0 | 0 to 0 | >0.999 |
| 24 h | 0 (0–2) | [0–0] | 0 | 0 to 0 | >0.999 |

**Notes.**

Scores were obtained at baseline, 15 min after sedation with acepromazine-buprenorphine, and between 0.5 to 24 h after extubation. Scores over time were compared with baseline using Friedman test and Dunn's multiple comparisons test. Sedation scores are presented as median (range) and interquartile range. Median of the differences (in comparison with baseline) and their 95% confidence interval (CI) are presented.

*Significantly different from baseline ($p < 0.05$).

**Table 2  Mean ± SD Feline Grimace Scale scores in real-time (FGS-RT) in cats undergoing ovariohysterectomy.**

| Time-point | FGS-RT scores Mean ± SD | Mean difference | 95% CI of the differences | *p* value |
| --- | --- | --- | --- | --- |
| Baseline | 0.16 ± 0.13 | – | – | – |
| 15 min | 0.2 ± 0.15 | 0.04 | 0.0045 to 0.0755 | 0.3847 |
| 0.5 h | 0.21 ± 0.14 | 0.047 | 0.0028 to 0.0920 | 0.2624 |
| 1 h | 0.26 ± 0.17* | 0.097 | 0.0462 to 0.1480 | 0.0003 |
| 2 h | 0.29 ± 0.2* | 0.132 | 0.0824 to 0.1820 | <0.0001 |
| 3 h | 0.25 ± 0.18* | 0.092 | 0.0434 to 0.1410 | 0.0003 |
| 4 h | 0.26 ± 0.16* | 0.106 | 0.0605 to 0.1520 | <0.0001 |
| 6 h | 0.23 ± 0.18* | 0.075 | 0.0277 to 0.1230 | 0.0095 |
| 8 h | 0.26 ± 0.14* | 0.097 | 0.0454 to 0.1480 | 0.0001 |
| 12 h | 0.22 ± 0.14* | 0.059 | 0.0096 to 0.1090 | 0.0316 |
| 24 h | 0.16 ± 0.12 | <0.0001 | −0.0434 to 0.0434 | 0.9998 |

**Notes.**

Scores were obtained at baseline, 15 min after sedation with acepromazine-buprenorphine, and between 0.5 to 24 h after extubation. Scores over time were compared with baseline using linear mixed models. The mean differences (in comparison with baseline) and their 95% confidence interval (CI) are presented.

*Significantly different from baseline ($p < 0.05$).

## Influence of sedation and OVH on individual action units

Sedation did not influence final FGS scores (both -RT and -IMG); however, orbital tightening scores obtained in real-time 15 min after sedation were higher than baseline ($p = 0.026$). This effect was not observed with any other action unit (ear position: $p = 0.72$; muzzle tension: $p = 0.17$; whiskers change: $p = 0.58$; head position: $p = 0.22$). Similarly, no significant differences in FGS-IMG scores were observed after sedation (ear position:

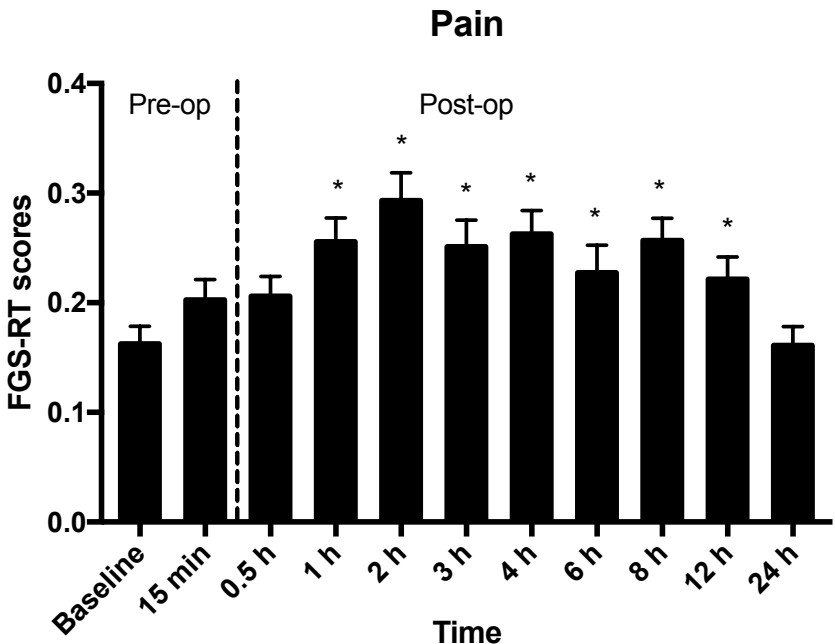

**Figure 4 Mean ± SEM Feline Grimace Scale scores in real-time (FGS-RT) in cats undergoing ovario-hysterectomy.** Scores were obtained pre- and postoperatively (Pre-op and Post-op, respectively) at baseline, 15 min after sedation with acepromazine-buprenorphine, and between 0.5 to 24 h after extubation. The average score of three minutes of observation of the cats' facial expression was considered the final FGS score per time-point. *Significantly different from baseline scores ($p < 0.05$).

$p = 0.1$; orbital tightening: $p = 0.32$; muzzle tension: $p = 0.14$; whiskers change: $p = 0.17$; head position: $p = 0.61$).

Ovariohysterectomy using a balanced anesthetic protocol did not influence any action unit scores obtained using either FGS-RT or FGS-IMG. All action units scores 24 h after extubation were not significantly different from baseline values (FGS-RT—ear position: $p = 0.076$; orbital tightening: $p = 0.35$; muzzle tension: $p = 0.72$; whiskers change: $p = 0.62$; head position: $p = 0.35$ and FGS-IMG—ear position: $p = 0.16$; orbital tightening: $p = 0.22$; muzzle tension: $p = 0.12$; whiskers change: $p = 0.1$; head position: $p = 0.93$).

### Rescue analgesia

Thirteen cats required rescue analgesia throughout the postoperative period. Mean ± SD scores before and after rescue analgesia were $0.47 ± 0.24$ and $0.21 ± 0.18$ (mean difference = $-0.266$, 95% CI of the differences: $-0.389$ to $-0.143$, $p = 0.0005$) respectively, using FGS-RT assessment; and $0.45 ± 0.19$ and $0.18 ± 0.17$ (mean difference = $-0.274$, 95% CI of the differences: $-0.481$ to $-0.0681$, $p = 0.0154$) respectively, using FGS-IMG assessment. Median (range) SF-UBCPS scores before and after analgesia were 7 (5–9) and 1 (0–3), respectively (median of differences: $-3$, 95% CI of the differences: $-4$ to $-3$, $p = 0.0005$).
## DISCUSSION

This study demonstrated the applicability of the Feline Grimace Scale in a clinical setting using real-time scoring. Minimal bias and narrow LOA were observed between scoring methods; the sedation protocol and surgery did not influence FGS scores and both FGS-RT and FGS-IMG detected responsiveness to analgesic administration.

The Bland & Altman for repeated measures method was used to verify the applicability of the FGS in real-time in a clinical setting. Real-time scoring using the FGS slightly overestimated image assessment (Bias$_{\text{final score}}$: $-0.057$). This could be explained by the clinical context of FGS-RT scoring and the three-minute observation of the cats' facial expression. Although the observer was focused on the action units to be scored, the body position and posture of the cat could be observed, and this might be another reason for the overestimation with the FGS-RT scores. Additionally, the FGS-IMG scores reflected the pain states using only one still image selected during that minute. It is possible that facial expressions might not be the exact same if an image was selected at the beginning or the end of each minute of observation, which may also explain the discrepancy observed with the FGS-RT when compared with FGS-IMG. Real-time scoring using the RGS was also demonstrated feasible with good agreement between real-time and image scoring (*Leung, Zhang & Pang, 2016*). In contrast to what has been observed in rats, real-time MGS scores were significantly lower than those obtained by image assessment in mice (*Miller & Leach, 2015*). In both rodent species, real-time scoring underestimated the scores obtained through image assessment. These differences might be related to the procedure for image capture. In rats, similar methodology was used; however, multiple intervals and punctual observations for RGS scoring methods were tested in real-time (*Leung, Zhang & Pang, 2016*). In mice, photographs were taken to obtain the MGS scores (*Miller & Leach, 2015*). In the latter study, the authors speculated that the use of photographs may have resulted in an artificial elevation of scores by capturing specific behaviors or movements (such as blinking), and that some difficulty in real-time scoring could be expected by their constant (and fast) level of activity. For the FGS, screenshots were taken from the videos when the cat was facing the camera. It is possible that the screenshots were obtained when the cat was paying attention to the surroundings (eyes open and ears facing forward) thus decreasing the resulting FGS-IMG score. Additionally, during FGS-RT scoring, the observer was aware of the timing (pre- or postoperatively). Although the non-blind nature of real-time assessment is inevitable and could represent a source of bias, it represents the method by which pain is evaluated in the clinical setting. Moreover, the good agreement between IMG and RT scores implies that the fact of the observer being aware of the timing did not introduce substantial bias.

There is no consensus for the classification of the LOA for pain scores with the same degree of standardization that exists for physiological variables (e.g., blood pressure, cardiac output, blood gas measurements, etc.) when designing measurement comparison studies (*Mantha et al., 2000*). As in the case of blood pressure monitors, the American National Standards of the Association for the Advancement of Medical Instrumentation recommends a limit for the mean difference of the paired measurements (bias) of $\pm 5$

mmHg or less between the test system and the comparison system (*White et al., 1993*). These criteria are not readily applicable for pain scores. Alternatively, the evaluation of the LOA in relation to the analgesic threshold has been proposed (*Leung, Zhang & Pang, 2016*). The LOA observed in the present study, although narrow (LOA$_{final\ score}$: $-0.351$ to 0.237), spans the analgesic threshold of 0.39 out of 1.0 previously determined for the FGS (*Evangelista et al., 2019*). This would mean that FGS-RT scores that are close to the analgesic threshold (slightly lower or higher than 0.39) should be interpreted with caution since this threshold is a suggestion for the administration of rescue analgesia based on the probability of being painful above that score. However, this should not be a major clinical problem if the final decision for giving additional analgesics rely on clinical judgement while taking in consideration the FGS score, context and disease, and the potential reassessment of the cat after a brief period.

Considering the small average discrepancy between methods for bias (with consistent variability for most of the FGS score range) and narrow LOA, the methods (FGS-IMG and FGS-RT) would probably provide similar assessments in the clinical setting. It should be noted; however, that the authors did not define the LOA *a priori* to state that real time and image assessments are interchangeable.

Although the assessment can be difficult in patients under the effects of anesthetics and sedatives, pain must be evaluated in a valid and reliable manner to ensure adequate treatment. For example, it is known that ketamine-based protocols confound pain scores in cats using the UNESP Botucatu Multidimensional Composite Pain Scale (*Buisman et al., 2016*). In this study, the premedication did not seem to affect FGS scores. Sedation scores increased after premedication with acepromazine-buprenorphine; however, it did not influence FGS scores using both methods of assessment (FGS-RT and FGS-IMG). Similarly, in mice and rats, the administration of buprenorphine alone had no impact on the MGS and RGS scores (*Miller et al., 2015*; *Leung, Zhang & Pang, 2016*). Even if the real-time evaluation (sedation scores and FGS-RT) might have been biased by the fact that the observer was aware of premedication, this type of bias is not present when using FGS-IMG scores. It is very unlikely that an observer could memorize all cat faces given the number of subjects, timepoints and the long delay between real-time assessment and evaluation of images after blinding and randomization. The action unit "Orbital tightening" was influenced by sedation using the FGS-RT. This effect might be related to the effect of acepromazine causing enophthalmos leading to protrusion of the third eyelid (*Hatch, Zahner & Booth, 1984*) and buprenorphine causing mydriasis (*Steagall, Monteiro-Steagall & Taylor, 2014*). This influence should be minimal in the clinical setting, since the final score was not affected by sedation and such changes were not detected using FGS-IMG.

The influence of OVH was determined by comparing the FGS scores 24 h after extubation with baseline values, in view of the long duration of action of the drugs used herein. Under this condition, OVH did not affect FGS scores (both FGS-RT and FGS-IMG). At this time-point we also would not expect high pain scores considering the minimal degree of surgical manipulation and the use of multimodal analgesia. Mean FGS-RT scores were significantly higher than baseline from 1 to 12 h after extubation; however, the mean

scores remained below the analgesic threshold ($\geq 0.39$ out of 1.0) and the majority of the cats did not require rescue analgesia. Drugs such as acepromazine, buprenorphine, bupivacaine and meloxicam are long-acting drugs, thus their effects would still be present during the postoperative period. In cats, increased sedation scores have been reported for up to 4 h after intramuscular administration of similar doses of acepromazine and buprenorphine (*Hunt et al., 2013*). Furthermore, the mean elimination half-life of bupivacaine was $4.8 \pm 2.7$ and $10.5 \pm 10.3$ h after intraperitoneal administration of bupivacaine alone and in combination with dexmedetomidine, respectively, in cats (*Benito et al., 2016*; *Benito et al., 2018*). Significant differences in pain or sedation scores, and the prevalence of rescue analgesia were not found between cats receiving intraperitoneal bupivacaine alone or in combination with dexmedetomidine (*Benito et al., 2019*). For this reason, the FGS scores considered in the present study were analyzed together. Meloxicam was administered at the 12 h time-point (except if a cat required rescue analgesia before it) and it has a serum half-life of approximately 24 h (*Lehr et al., 2010*). Even though the effect of other drugs may have worn off, the results observed at 24 h after extubation may have been influenced by the long duration of action of meloxicam.

Responsiveness to rescue analgesia, as part of construct validity testing, was previously assessed during the development and validation of the FGS using various analgesic protocols including different drugs, doses and routes of administration (*Evangelista et al., 2019*). In that study, FGS-IMG scores decreased after the administration of analgesics when compared with those at presentation (i.e., before interventional analgesia). In the present study, responsiveness was tested again since the study design included a standardized protocol for rescue analgesia and type of surgical stimulus (OVH). Additionally, responsiveness should also be assessed using FGS-RT. Both methods of pain assessment (FGS-RT and FGS-IMG) detected changes in pain scores corroborating our previous findings (*Evangelista et al., 2019*). Decreases in pain scores were also observed with the SF-UBCPS.

During real-time scoring the observer was present in front of the cage and was able to move around to look at the cat from different angles, while videos were recorded simultaneously from a single angle. The cats' faces were not always visible from the camera angle when screenshots were taken, thus explaining missing FGS-IMG scores in many cases. Indeed, it was not possible to obtain any image from 160 videos (approximately 40%) for the following reasons: movement of the cat either too close or far from the front of the cage and out of view; when cats were facing the back of the cage or hiding behind the litter box, sleeping, grooming, or when the image was blurred. Perhaps the number of excluded images could be reduced by using two cameras, placed on either side of the cage, as reported in mice (*Langford et al., 2010*), rats (*Sotocinal et al., 2011*) and horses (*Dalla Costa et al., 2014*). The effect of the presence of the observer in front of the cage while the videos were recorded was not evaluated in this study; however, it is currently being investigated. A recent study in mice showed that the presence of a male observer in the room reduced MGS scores, whereas the same was not observed with female observers. These differences in MGS scores are likely a result of stress-induced analgesia (*Sorge et al., 2014*). Similarly, the presence of a female observer did not interfere with RGS scores in

rats (*Leung, Zhang & Pang, 2016*). In the present study, a female observer performed pain assessments and it remains unknown if the gender of the observer influences FGS scores in cats.

Some limitations of this study include the use of a single type of surgical painful stimulus and model of acute pain (OVH). Effects of sedation were assessed 15 min after premedication, before the surgery and any other painful stimulus and only one protocol for premedication was studied. Therefore, our findings are still limited to OVH involving premedication and mild sedation with acepromazine-buprenorphine. It is not clear how other sedative protocols producing moderate to profound sedation may affect the FGS scores. In addition, it was not possible to evaluate the effects of anesthesia and surgery separately on FGS scores since we did not include sham and negative control groups undergoing general anesthesia without the surgical procedure, or surgery without the administration of analgesics (the latter for ethical reasons). This is a clinical study involving surgery that requires the administration of anesthetics; therefore, it is an intrinsic limitation of the methodology. The effects of general anesthesia alone on pain scores were assessed in rats and horses. Higher RGS scores were observed in rats in the immediate time period, approximately 20 min after the discontinuation of short duration isoflurane anesthesia (*Miller, Golledge & Leach, 2016*); however, Horse Grimace Scale scores were unchanged 8 h after general anesthesia without the application of a nociceptive stimulus (*Dalla Costa et al., 2014*).

## CONCLUSIONS

Real-time scoring using the FGS is feasible, although it slightly overestimates image assessment. The minimal bias and narrow limits of agreement between FGS-RT and FGS-IMG suggest minimal clinical impact. Sedation with acepromazine-buprenorphine and ovariohysterectomy using a multimodal analgesia and a balanced anesthetic protocol did not influence the FGS scores. Responsiveness to analgesic administration was detected both with FGS-RT and FGS-IMG.

### Funding

This study was funded by an unrestricted grant provided by Zoetis. Funding was also provided by the Companion Animals Health Fund from the Faculty of Veterinary Medicine of the Université de Montréal, supported by Zoetis and a donation by Ms. Valeria Rosenbloom and M. Mike Rosenbloom. The funders had no role in study design, data collection and analysis, decision to publish, or preparation of the manuscript.

### Grant Disclosures

The following grant information was disclosed by the authors:
Companion Animals Health Fund from the Faculty of Veterinary Medicine of the Université de Montréal.
Unrestricted grant by Zoetis.

## Competing Interests

Beatriz Monteiro and Paulo Steagall have provided consultancy services for Zoetis. This does not alter the authors' adherence to PeerJ policies on sharing data and materials.

## Author Contributions

- Marina C. Evangelista conceived and designed the experiments, performed the experiments, analyzed the data, prepared figures and/or tables, authored or reviewed drafts of the paper, and approved the final draft.
- Javier Benito, Beatriz P. Monteiro, Ryota Watanabe and Graeme M. Doodnaught performed the experiments, authored or reviewed drafts of the paper, and approved the final draft.
- Daniel S.J. Pang analyzed the data, authored or reviewed drafts of the paper, and approved the final draft.
- Paulo V. Steagall conceived and designed the experiments, performed the experiments, analyzed the data, authored or reviewed drafts of the paper, and approved the final draft.

## Animal Ethics

The following information was supplied relating to ethical approvals (i.e., approving body and any reference numbers):

The protocol was approved by the local animal care committee, Comité d'éthique de l'utilisation des animaux—Université de Montréal (protocol number 18-Rech-1825).

## Data Availability

Raw data are available as Supplemental File.

## Supplemental Information

Supplemental information for this article can be found online at http://dx.doi.org/10.7717/peerj.8967#supplemental-information.

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
