# Peer review of "Clinical applicability of the Feline Grimace Scale: real-time versus image scoring and the influence of sedation and surgery"

_PeerJ, doi:10.7717/peerj.8967_

## Round 0.1 · original submission · Major Revisions

Please carefully review the suggestions made by both reviewer number one and two. Also there is some questions as to a paper you are referencing to as in press or submitted. Please take careful concern to make sure this is properly cited.

Reviewer 1 ·

Basic reporting

Generally well executed. See specific section within general comments for grammatical errors noted. Mention of previous cat pain assessment methods, particularly those focusing on facial expressions would be useful in order to supply sufficent background/context.

Experimental design

Some limitations regarding reproducibility of methods and potential design flaws due to coding bias – see comments in following section

Validity of the findings

During RT scoring, it seems that the coder may have had knowledge of the general posture and behaviour of the cat, their level of sedation and the relative time point of their assessment (i.e. pre or post op and time since surgery), as well as their SF-UBCPS scores. Important controls to add here would be to assess inter-coder reliability of the still images that were coded, using a properly blinded coder. The second would be to get the blinded coder to code the FGS from the videos, using the same methods as FGS-RT scoring.

Additional comments

This is a worthwhile contribution to the field of clinical pain assessment based on facial expressions in cats. However, it makes it difficult to fully evaluate its merit when the efficacy of the main tool of measurement being tested (FGS) has yet to be published, and only very limited detail about the FGS and how it was performed is provided. As such the methods are potentially not easily reproduceable in their current state. I would recommend either delaying submission until the paper referencing the FGS is published, or include sufficient detail in the methods to support the FGS scoring method.

Additionally, there is a potentially substantial confound in the way that FGS coding was performed which could call into the question the validity of the main assumptions being made within the paper. The fact that there are potentially substantial biases present during RT coding, that the same person then coded the still images, and that the authors found a difference between RT and IMG scores limits the ability of this paper to properly demonstrate the clinical applicability of the FGS in real time. These issues should be addressed - please see previous comments for suggestions.

Specific section comments:

Abstract:
“Cats were video-recorded simultaneously at baseline, 15 minutes after sedation, and at 2, 12, and 24 hours after extubation for subsequent image assessment (FGS-IMG) performed six months later” – the last part of this sentence could do with a little rewording to explain the relevance of the 6 month image assessment

Introduction:

Ln 66: It would be really useful here to provide a brief example of the confounding effects found

Ln 69: This statement is not true entirely true – see Finka et al 2019

Ln 70: ‘there’ not ‘they’

Your first hypothesis suggests a possible design flaw due to the non-blind nature of image scoring

Your second hypothesis is reasonable, although it is not fully supported by the literature discussed in the introduction. If you are going to test a specific hypothesis in this way, I would recommend providing sufficient evidence in advance to justify this

Methods:
The method of FGS scoring and video recording of cats needs further explanation to enable it to be easily replicated. It is not sufficient to state that methods followed those detailed elsewhere and to reference a paper that is currently in submission.
Please state whether the person preforming the RT FGS was blinded to the cat’s SF-UBCPS scores and relative scoring time point at which scoring took place as these may otherwise act as potential confounds.
Please also include information detailing the way scoring took place (i.e. was the scorer told only to look at the cat’s face, or were they able to observe other features of the cat and also perform other types of observation or clinical assessment on the cat?)
The linear mixed models could do with further explanation – e.g. how may separate models were performed, and the response and fixed effects/explanatory variables in each case. I’m struggling to understand how the outputs in table 2 relate to the LMMs as currently described. I would also suggest you briefly mention how your data met the criteria/assumptions for each of the statistical tests you performed in support of their selection.
Ln 211 – is there a suitable reference to support the conclusion of 'narrow LOAs' as stated?
Ln 222: Would it not make sense to analyse and report the differences in RT and IMG FGS at different time points compared to baseline in the same way here?
Ln 242 – surely (as supported by your results) surgery did influence action units obtained via FGS in line with induction of post-op pain, although I’m not sure this is the point you are trying to make here?

Discussion:
Ln 227: It would be worth to speculate the possible reasons for the over estimation of real-time scoring in one species and the underestimation in another, rather than just state that this is the case.
Paragraph 266: Is it also not possible that the difference in scores or ‘over estimation’ may be due to the RT scoring method (i.e. the scorer is not blind to the cat’s current condition, they have examined the cat, performed other composite pain assessment test and can observe the whole of the cat and their demeanour prior to FGS scoring?)
Ln 279: It’s great that you touch upon the lack of consensus regarding LOAs for pain scores and its clinical relevance to analgesic intervention
Ln 293: taken not taking
Ln 305: If the FGS scorer was not blinded to the sedation score of the cat, can you properly conclude that sedation scores did not affect FGS?

Reviewer 2 ·

Basic reporting

No comment

Experimental design

The article meets all of the adjacent criteria.

Validity of the findings

No comment

Additional comments

The authors describe a convincing test of the applicability of the FGS in a clinical setting, comparing both real-time and image-based assessment of an impressive number of cats. The results and conclusion appear to support a previous 'reported' study ( Evangelista et al); but unfortunately this is not yet available. In numerous places this is described as 'submitted', but according to the conclusions section the article is 'in press'. Which is it ? Otherwise, in most respects the the article is very good one, but there are occasions when greater clarity might improve it. In particular, the reviewer was unclear about the timing of recording of each action unit of the FGS. Line 137. It is not clear what is meant by 'one score per minute of observation'. Does this mean all 5 FAU's were scored during minutes 1, 2 and 3? If so, please say this another way. Likewise, line 163 says one image per minute was selected does this mean the clip was paused at 1, 2 and 3 minutes, and the image clipped? I find it hard to believe that selecting these times would have avoided times when cats were not looking at the camera. The is no doubt a simple explanation of this given greater clarity in the description. Again, on line 168, 'the average score of the 3 minutes' is unclear.
Again, I do not see the range of selected images (line 204) could include 0 or have a maximum of 20? Why did you not choose the minimum acceptable number for each cat and select from these randomly.? Perhaps this '0' value recognizes that 160 videos were unusable. This is a worryingly high number, and that fact should be emphasized in the discussion as a general problem in off-line data analysis using any type of grimace scale.
Introduction line 76 and Discussion line 263. It is a little confusing when you say surgery did not influence FGS scores, when presumably the whole point of the investigation was determine that it did; ie caused pain that can be detected with the FGS. Similarly, In the introduction why would you hypothesize that FGS scores would not be elevated by 24 hours compared with baseline? This suggests you thought it unlikely that the effects of OVH would not extend beyond that time? Why? - This arises again on line 320 where you say OVH did not affect FGS scores, when clearly it did. Please clarify these statements or re-state the original hypothesis on this aspect if the reviewer has misunderstood. Line 274, in contrast to what?
Overall, this article is ethically sound and well presented and the outcomes seems clear and worthy of publication after resolution of these relatively small issues. However, Please state whether all cats were successfully re-homed or what proportion had to be euthanased, if any.

Other minor points: Introduction, line line 61, this is usually called the Mouse Grimace Scale. Line 129, what is meant by weight was redistributed to the other action units? I am assuming this means if only 4 units were scored, then the combined total was divided by 4 and not 5. Please clarify.

---

## Round 0.2 · Major Revisions

Please use Reviewer number three's recommendations as a template for revision (Reviewer 2 was unavailable to re-review). However, please make sure you are able to adequately address the ethics review concern brought up by the reviewer, such that you can document that your study was reviewed and approved by an Institutional Review Board or like applicable entity. Without such approval or exemption and related documentation it may not be possible to move forward with any publication process.

Reviewer 1 ·

Basic reporting

no comment

Experimental design

no comment

Validity of the findings

no comment

Additional comments

Thank you for taking the time to address the various comments - the clarity of the manuscript and its methodology is improved.

A few outstanding grammatical issues:

Ln 29 “Cats were video-recorded simultaneously at baseline, 15 minutes after sedation, and at 2, 6, 12, and 24 hours after extubation for subsequent image assessment (FGS-IMG) performed six months later by the same observer” – suggest adding in ", which was" before the word 'performed' for ease of reading

Ln 254 Ovariohysterectomy did not influence any action unit scores obtained using either FGS-RT or FGS-IMG. - this sentence still needs attending to – wasn’t the whole point of your study to show exactly this?! I think the point you want to make is that after pain was abated, the surgical intervention in itself didn’t appear to affect FGS…this issue of wording also still needs addressing in the discussion

Reviewer 3 ·

Basic reporting

Clear language and narrative to the paper. Some minor errors in grammar and syntax. Check throughout that there is only one space between words and not two. Additionally, there should be either a full line space between paragraphs or they need have an indentation on the first line to make it clear that a new paragraph has been started. When using ‘however’ to link two parts of a sentence their needs to be a semi-colon before and a comma after the however.

Relevant materials included and structure appears to follow the PeerJ standards. Intro and background missing some key references – see section below for more details.

Experimental design

Within scope of journal. Research question and hypothesis could be clearer and better justified. Some basic details missing from methods making replication difficult. The basis of the paper (FGS) is from a paper that seems to be referred to as submitted and not yet published. This should be clarified as to the validity of the scale used.

Validity of the findings

If based on non-published FGS, then validity can be questions – see point above. Some areas of bias and lack of sufficient methods can call into question the validity of the findings. Some statements/conclusions over stated/lack sufficient evidence.

Additional comments

Comments for Author
It is good to see this type of study having been carried out as they are much needed to encourage the use and development of these scales. You have made good use of the resources of another study which helps to reduce the number of animals used. You find it useful to look at Descovich et al., and McLennan et al.’s review papers on the facial expression development and use. In particular the later discusses the use of facial expression scales in clinical practice as well as highlights some additional consideration that this paper has not yet addressed.

Abstract
Clear abstract. Line 23 should have a comma after ‘assessment’.

Introduction
It is mentioned in the discussion (lines 368-370) that responsiveness is being measured again, in addition to it already being carried out in Evangelista et al., submitted. Should this not be made clear in the introduction and so the changing of the aims and hypothesis to replicate what they have done?

The end of your introduction highlights the fact that the effect of sedation and OVH itself has not been studied yet with regards it affects in FGS scores and if there are potential limitations to the use of FGS in clinical practice. Your aims I do not believe really address this point, nor do the results or experimental design really address the aims of the study. A rephrasing and better leading justification is needed here to make it clearer what this study is about and why.

Line 50 – should read ‘under recognized’.
Line 52 – remove additional spacing between reference and ‘Facial expression…’.
Paragraphs need clear spacing or indentation.
Line 53 – mention ‘mammals’ but limited references to cats. Given how many other scales are now present, this is too limited and I suggest you put a few more other papers in the references list to demonstrate this.

Methods
Why only females included – I know OVH, but does this bias the results to just females? There are differences between strains of mice, let alone how males and females react to situations and potential express their affective state.

The cats were from a local shelter, yet you state permission for inclusion was gained from, but it is not clear who was responsible for these cats and whether they were really in the right to give permission. In addition, the ethics was given to the other study, and there is not mention of this particular study (although using another experiment) was assessed. This study has its own ethical concerns that are separate from the other study and so should have been assessed. The main study that the present stemmed from also had a different aim and therefore this should have been addressed within this study as well to show how these two treatments would have also affected the results. I would like to know how this was controlled for and what effect this may have had on the results.

Lines 112-115 is unclear and needs rephrasing.
Line 129 – cats receiving a score of >2 on the SF-UBCPS not included – why? No justification given for this.
Lines 134-138 describe a different method for calculating the FGS score compared to that used previously in these types of studies – what is the justification for this method? Was this the method used for RT or IMG or both?

How are the effects of sedation and pain being separated?

More detail required on how RT scores at baseline were obtained. Line 143 states being ‘undisturbed’, but if an observer is there to carry out the RT assessment, they are not really undisturbed.

Line 143 also states ‘during each minute’, but exactly how in each of these minutes’ assessment was carried out is unclear. It would not be possible to replicate this study with the current lack of details on exactly how the study was conducted.

In addition, more details on when baseline occurred and why animals were included in the study – what was there reason for being there and what previous ailments did these have. More clarity on this is required.

Why were there different times in which FGS was assessed between RT and IMG – why not keep them the same?

When were animals assessed for the need for additional rescue analgesia? What conditions needed to be met to reach this decision? Why was meloxicam given at 12h for each cat? How was the effect of this considered in the results?

Line 166 states videos were not taken if previously had been spayed – is this not what the study was looking at - OVH? Was there not a possibility of sham and surgical, or was it not known before if these animals had or had not been spayed previously? Yet, the videos for baseline and 15 mins were included – how did you get then an equal study design to assess the intra-animal changes in facial expressions?

Line 169 – screen shots taken’ – no detail of exactly how they were chosen within each of the minute e.g. frame by frame, after the first 15secs, or just the very first clear image.

No mention of controls used or match pairs.

Good to have six months in between RT and IMG assessment – any possibility of observer fatigue of training effect? Any intra-observer testing carried out between RT and IMG timings?

Was the RT observer blind to the post and pre-op state of animals?

Why was SAS and GraphPad prism chosen? Why not R, which can handle this type of complex data much better?

A brief explanation of Bland and Altman would be useful here to justify its use.

Results
It is not clear why there is a possibility of 0-20 possible videos. This was previously pointed out by another reviewer and do not feel it has been sufficiently addressed in the paper so that it is clear for any reader to understand.

135 images assessed each day – what about observer fatigue? Were any intra-observer reliability tests carried out from the first to last image etc?

You state OVH did not influence any action unit but figure 4 shows a difference in scores pre and postoperatively? So, I’m unsure how you can have these two things? How are you separating this out? How can total pain be different, but the FGS not affected? Was this an effect of the analgesia? What were the two different treatments like and were they different?

I’m struggling to understand the data in tables 1 and 2, especially table 1 that is titled scores pre and post, yet there is only one score?

Figure 3 shows quite a range in agreement scores given that the actual scores range from 0-1. How would this change if you took the total FGS score and not a fraction of it?

Discussion
Line 288 – avoid ‘proven’. Science proves nothing – it can only demonstrate/show.

Line 384 – How did you eliminate the effect if the observer on the RT score? If you couldn’t, you need to take this into account when concluding what your results truly show.

Line 388 – needs expansion and explaining why you consider the differences are likely to be stress-induced analgesia and not the effects of different regimes of treatment, etc.

Line 397-398 – It is good that you recognise you cannot separate out the pain and analgesic effect, but you also need to consider this in the rest of your discussion.

Conclusion
You state that the IMG assessment over-estimates the RT score, but what if it were that the RT underestimates. You need to be careful with your phrasing if you do not have the evidence to back up the statement.

---

## Round 0.3 · Minor Revisions

Thank you for your patience and persistence in preparing this manuscript for publication. Please make the remaining revisions as suggested by reviewer number three.

Reviewer 3 ·

Basic reporting

Minor changes suggested for change. See detailed comments below.

Experimental design

Minor changes suggested for change. See detailed comments below.

Validity of the findings

Minor changes suggested for change. See detailed comments below.

Additional comments

Thank you to authors for their detailed responses to my previous comments. The changes and responses to my questions have made the much clearer. It is now a much more concise and clear paper and I commend the authors for engaging so well in the process. There are a few minor changes I would recommend:

- Make clear that the effects of sedation and pain being separated has been fully covered in the discussion as an intrinsic limitation to the study.
- Again in the discussion, make clear that the lack difference between the RT and IMG scores implies/ suggests no bias from the observer
- Check that objective 3 and hypothesis 3 match clearly
Line 51: under treated rather than undertreated
Line 78: should read “..OVH (Finka et al., 2019). It is not known…”
Line 120: It wasn’t clear what the two treatments are.
Lines 131-133: Adding in a justification for this would be good, e.g. why assess sedation.
Line 158: Did you carry out one-zero sampling or instantaneous sampling? Adding this in would make this clearer.
Line 175: State why you needed to give meloxicam at 12 hour, as you made clear in your replies from my previous review.
Line 243-244: Delete this last sentence.
Line 248: add in “..to avoid observer fatigue.”
Line 310-311: I’m still not comfortable with this wording here. I think your phrasing in the conclusion that RT scoring over-estimates image assessment is clearer. Consider rephrasing this.
Line 376: Remove the ‘However’ at the beginning of the sentence so that it starts “This influence should be …”.

---

## Round 0.4 · accepted · Accept

Thank you for your patience with the revision process. As you are well aware, while peer review can at times seem tedious, I am a firm believer in the process and thank you for your professional and scholarly persistence.